# Transcriptome Analysis of Ochratoxin A-Induced Apoptosis in Differentiated Caco-2 Cells

**DOI:** 10.3390/toxins12010023

**Published:** 2019-12-31

**Authors:** Xue Yang, Yanan Gao, Qiaoyan Yan, Xiaoyu Bao, Shengguo Zhao, Jiaqi Wang, Nan Zheng

**Affiliations:** 1Key Laboratory of Quality & Safety Control for Milk and Dairy Products of Ministry of Agriculture and Rural Affairs, Institute of Animal Sciences, Chinese Academy of Agricultural Sciences, Beijing 100193, China; 82101182216@caas.cn (X.Y.); gyn758521@126.com (Y.G.); yeqiaoyan@caas.cn (Q.Y.); xbao@ualberta.ca (X.B.); zhaoshengguo@caas.cn (S.Z.); wangjiaqi@caas.cn (J.W.); 2Laboratory of Quality and Safety Risk Assessment for Dairy Products of Ministry of Agriculture and Rural Affairs, Institute of Animal Sciences, Chinese Academy of Agricultural Sciences, Beijing 100193, China; 3Milk and Dairy Product Inspection Center of Ministry of Agriculture and Rural Affairs, Institute of Animal Sciences, Chinese Academy of Agricultural Sciences, Beijing 100193, China; 4State Key Laboratory of Animal Nutrition, Institute of Animal Sciences, Chinese Academy of Agricultural Sciences, Beijing 100193, China

**Keywords:** ochratoxin A, differentiated Caco-2 cells, cell apoptosis, transcriptome analysis

## Abstract

Ochratoxin A (OTA), an important mycotoxin that occurs in food and animal feed, has aroused widespread concern in recent years. Previous studies have indicated that OTA causes nephrotoxicity, hepatotoxicity, genotoxicity, immunotoxicity, cytotoxicity, and neurotoxicity. The intestinal toxicity of OTA has gradually become a focus of research, but the mechanisms underlying this toxicity have not been described. Here, differentiated Caco-2 cells were incubated for 48 h with different concentrations of OTA and transcriptome analysis was used to estimate damage to the intestinal barrier. Gene expression profiling was used to compare the characteristics of differentially expressed genes (DEGs). There were altogether 10,090 DEGs, mainly clustered into two downregulation patterns. The Search Tool for Retrieval of Interacting Genes (STRING), which was used to analyze the protein–protein interaction network, indicated that 24 key enzymes were mostly responsible for regulating cell apoptosis. Quantitative reverse transcription-polymerase chain reaction (qRT-PCR) analysis was used to validate eight genes, three of which were key genes (*CASP3*, *CDC25B*, and *EGR1*). The results indicated that OTA dose-dependently induces apoptosis in differentiated Caco-2 cells. Transcriptome analysis showed that the impairment of intestinal function caused by OTA might be partly attributed to apoptosis, which is probably associated with downregulation of murine double minute 2 (MDM2) expression and upregulation of Noxa and caspase 3 (CASP3) expression. This study has highlighted the intestinal toxicity of OTA and provided a genome-wide view of biological responses, which provides a theoretical basis for enterotoxicity and should be useful in establishing a maximum residue limit for OTA.

## 1. Introduction

Ochratoxin A (OTA) is a fungal secondary metabolite produced by certain *Penicillium* and *Aspergillus* species, including *Penicillium verrucosum*, *Aspergillus ochraceus*, and *Aspergillus niger* [1]. OTA was first isolated from *A. ochraceus* in 1965 and was found to contaminate the food chain worldwide [2]. It is widely found in various grains and vegetables [3,4,5], as well as in food products of animal origin, such as meat, eggs, and milk [6,7,8,9]. Milk, which has high bioavailability and is an abundant source of nutrients, is widely recognized to be an important component of the human diet. As the consumption of milk has increased over recent years, the mycotoxins found in milk have received increasing attention. A provisional tolerable weekly intake of 100 ng/kg.bw/week has been established for OTA by the Joint FAO/WHO Expert Committee on Food Additives (JECFA) [10], JECFA, 2001), although a maximum residue limit (MRL) for OTA in milk has not been agreed upon internationally. A study in Italy detected OTA concentrations of 70–110 ng/L in organic milk [11]. In China, Huang et al. [12] measured levels of OTA in raw, powdered, and liquid cow’s milk, and found mean concentrations of 56.7, 27.0, and 26.8 ng/kg, respectively. In Sudan, the level of OTA in a contaminated milk sample was as high as 2730 ng/L [13]. OTA has been classified in Group 2B (possible carcinogens in humans) by The International Agency for Research on Cancer, because of evidence of carcinogenicity in animals, but not in humans [14].

The gastrointestinal tract (GIT) is essential for human health and provides a barrier between the external environment and the tightly regulated internal environment [15]. The GIT can be exposed to numerous contaminated foods and high doses of some mycotoxins [16]. Early studies on OTA focused mainly on the diversity of toxic effects in different animal species [17,18,19]. Recent studies, however, have reported the toxic effect of OTA on the intestine [20,21]. OTA-induced intestinal damage has been reported in both animals and in vitro intestinal models [22,23]. It has been shown to damage the intestinal epithelium in chickens and rats [24,25,26] and also shows toxicity in intestinal epithelial cells, including a porcine intestinal cell line (IPEC-J2) and human intestinal epithelial lines (HT-29-D4 cells and Caco-2 cells) [20,27,28]. Previously published studies have shown that cell apoptosis is one of the ways by which OTA exerts intestinal toxicity [20,26,29]. Wang et al. [20] suggested that apoptosis induced by OTA may play a major role in the intestinal toxicity of this mycotoxin, and Bouaziz et al. [30] also suggested that OTA causes toxicity through apoptosis. OTA has been shown to induce apoptosis in different cell lines [31,32,33,34,35], which may be one of the main cellular mechanisms underlying the toxic effects. However, the molecular mechanisms responsible for cell apoptosis, which leads to intestinal toxicity, are still inadequately understood. It is, therefore, important to investigate the mechanism of apoptosis of intestinal epithelial cells following exposure to OTA. In the present study, we used differentiated Caco-2 cells as they can form polarized apical/mucosal and basolateral/serosal membranes that are similar to those formed by epithelial cells in the small intestine [36]. Moreover, it has been acknowledged by the Food and Drug Administration that differentiated Caco-2 cells are a suitable model for evaluating the impact of toxins on intestinal barrier function [37,38].

With large-scale transcription approaches, a comprehensive overview, provided by high throughput data, can easily reveal biological pathways and processes that have not been found before [39]. Using whole-genome transcriptome profiling, RNA sequencing (RNA-seq), an unbiased sequencing tool, has been used to detect changes of gene expression in tissue samples or cells [40,41]. The aim of this study was to investigate the mechanism of OTA-induced apoptosis in differentiated Caco-2 cells and to use RNA-seq technology to evaluate changes in gene expression and profile to clarify the mechanism of OTA-induced apoptosis in differentiated Caco-2 cells.

## 2. Results

### 2.1. OTA Induces Apoptosis in Differentiated Caco-2 Cells in a Dose-Dependent Manner

After treatment with OTA (0.0005, 0.005, and 4 μg/mL) for 48 h, Annexin V-PI dual staining and flow cytometry were used to detect apoptosis of differentiated Caco-2 cells (Figure 1a). As depicted in the histogram (Figure 1b), the percentage of apoptotic cells was increased by treatment for 48 h with 0.0005 μg/mL (2.9% ± 0.67%), 0.005 μg/mL (5.23% ± 0.21%), and 4 μg/mL (20.9% ± 2.49%) OTA, compared with the control (2.3% ± 0.08%). The percentage of cell apoptosis was significantly increased in differentiated Caco-2 cells (*p* < 0.05) when the concentration of OTA was 4 μg/mL (Figure 1b). The number of living cells in the Q3 regions of the flow cytometry plots also decreased as the concentration of OTA increased (Figure 1a).

### 2.2. Effect of OTA on Gene Expression Patterns

There were 18,654 unigene annotations in the control group (56.56% of all the 32,938 reference unigene sequences) and 18,475 (56.56%), 18,898 (57.30%), and 19,853 (55.61%) unigene annotations in the 0.0005, 0.005, and 4 μg/mL OTA groups., respectively. Using a *p*-value < 0.05 and a 2-fold change (FC) as the conditions for discrimination, we identified 503, 2139, and 9402 differentially expressed genes (DEGs) when the cells were incubated for 48 h with 0.0005, 0.005, and 4 μg/mL OTA, respectively. At these three concentrations of OTA, 35, 258, and 1437 genes were upregulated and 468, 1881, and 7965 genes were downregulated, respectively (Figure 2). At all concentrations of OTA, downregulation of genes was the main trend (93%, 88%, and 85% of the total DEGs, respectively). The number of DEGs in differentiated Caco-2 cells increased with increasing OTA concentration. Using a Venn diagram, we found that 3 upregulated DEGs and 169 downregulated DEGs were commonly modulated by all three concentrations of OTA (Figure 3a,b). Compared with the control, there were 257 overlapping DEGs between the 0.0005 and 0.005 μg/mL OTA groups and 1532 overlapping DEGs between the 0.005 and 4 μg/mL groups (Appendix A), showing that the common DEGs were increased in a dose-dependent manner. To assess the expression patterns of mRNAs at different concentrations of OTA, we used heatmaps to analyze the overall transcriptome differences (Figure 3c). The same differently expressed transcripts were present in three separate runs of the control group (CTL1, CTL2, CTL3), the 0.0005 μg/mL treatment group (0.0005-1,2,3), the 0.005 μg/mL treatment group (0.005-1,2,3), and the 4 μg/mL treatment group (4-1,2,3) (Figure 3c). The heatmap (Figure 3c) shows accurate repeatability and high reliability.

### 2.3. Gene Ontology (GO) Annotation and KEGG Enrichment Analysis of DEGs

Using the FC and *p*-value thresholds described in the Methods section above, we obtained 10,090 DEGs (compared with the control, the sum of DEGs of the OTA treatment at different concentrations), which could be clustered into eight profiles using the short time-series expression miner to obtain dynamic expression patterns of the DEGs. Among these eight profiles, two classic downregulated profiles (profiles 0 and 3) were significantly enriched (Appendix A). A total of 7199 DEGs were mainly clustered into these two downregulated profiles, which contained 1883 and 5226 genes, respectively (Appendix A). To discover the functions of the DEGs and the associated biological processes altered by OTA treatment of differentiated Caco-2 cells, we carried out a gene ontology (GO) (http://www.geneontology.org/) enrichment analysis, in which the DEGs were divided into three independent GO categories. Single DEGs could be annotated to more than two GO terms and the most enriched GO terms are displayed in Appendix A. The three main categories of the GO classification were assessed for enriched genes.

All the DEGs were mapped in the Kyoto Encyclopedia of Genes and Genomes (KEGG, https://www.kegg.jp) database to detect the response pathways altered by treatment with OTA. Of the DEGs that could be annotated to the KEGG pathway, most were associated with metabolism and signal transduction pathways. In profile 0, the enriched pathways were pyrimidine metabolism, the TNF signaling pathway, ribosome biogenesis in eukaryotes, and the Wnt signaling pathway. In profile 3, the enriched pathways were endocytosis, cell cycle, ubiquitin-mediated proteolysis, pyrimidine metabolism, and AMPK signaling (Figure 4).

### 2.4. Key Pathways and DEGs Related to Cell Apoptosis

Using the GO annotation and KEGG analysis, we can select the key pathways that are associated with the toxic effects of OTA in differentiated Caco-2 cells. DEGs from the key significantly enriched pathways, particularly those shared between the different concentrations of OTA, may be thought of as the key gene expression regulators that respond to treatment with OTA. These 10 key pathways, together with some key genes that participate in or regulate the cell cycle and cell apoptosis, are shown in Table 1.

As shown in Figure 5, the 50 DEGs that occurred frequently in these key pathways, or participated in individual key pathways, were combined and employed to construct a protein–protein interaction (PPI) network using STRING. Cytoscape 3.1 was then used for further filtering (Edge score > 0.8). The 10 key pathways and 24 key DEGs associated with cell apoptosis were identified mainly from those shared between the different concentrations of OTA in the GO and KEGG analysis, and these key pathways were largely associated with the regulation of cell apoptosis and the cell cycle. According to the degree of connectivity of each node, we selected the following 24 key enzymes: murine double minute 2 (MDM2), v-akt murine thymoma viral oncogene homolog 1 (AKT1), tumor protein p53 (TP53), caspase 3 (CASP3), caspase 9 (CASP9), hras proto-oncogene gtpase (HRAS), mechanistic target of rapamycin kinase (MTOR), epidermal growth factor receptor (EGFR), bcl-2-like protein 1 (BCL2L1), tumor necrosis factor (TNF), atm serine/threonine kinase (ATM), cytochrome C somatic (CYCs), mitogen-activated protein kinase 1 and 14 (MAPK1, MAPK14), ribosomal protein s6 kinase B1 (RPS6KB1), x-linked inhibitor of apoptosis (XIAP), nuclear factor kappa B subunit 1 (NFKB1), cyclin B1 (CCNB1), E1A binding protein p300 (EP300), inhibitor of nuclear factor kappa B kinase subunit beta (IKBKB), TSC complex subunit 2 (TSC2), phosphatidylinositol-4,5-bisphosphate 3-kinase catalytic subunit alpha (PIK3CA), cyclin dependent kinase 1 (CDK1), and forkhead box O1 (FOXO1).

### 2.5. Validation of RNA-Seq Results by qRT-PCR

Quantitative reverse transcription-polymerase chain reaction (qRT-PCR) analysis was performed to validate the high throughput data. We determined the expression of the following eight randomly selected genes: caspase 3 (CASP3), C–X–C motif chemokine ligand 2 (CXCL2), cell division cycle 25B (CDC25B), early growth response 1 (EGR1), FRY like transcription coactivator (FRYL), H2B histone family members (H2BFS), sedoheptulokinase (SHPK), and transcription factor EC (TFEC). Expression levels of CDC25B, SHPK, ERG1, FRYL, and TFEC significantly decreased after OTA treatment. In contrast, expression of CASP3, CXCL2, and H2BFS significantly increased (Figure 6a). These expression patterns were in good agreement with the data obtained by RNA-Seq, confirming the reliability of the two methods. There was a strong correlation (r = 0.803) between RNA-seq and qRT-PCR (Figure 6b). The protein levels of two randomly selected genes, MDM2 and CASP3, were measured by Western blotting. MDM2 levels significantly decreased and CASP3 levels increased after treatment with OTA (Figure 7).

## 3. Discussion

OTA has deleterious effects on humans and animals, resulting in worldwide illness and economic losses. Exposure to OTA occurs from ingestion of contaminated food and feed [42]. The small intestine is the main site of OTA absorption, with the largest absorption in the proximal jejunum [43]. Because of their location and function, intestinal epithelial cells are a potential target for the toxic effects of OTA. In the present study, flow cytometry showed that the number of apoptotic cells increased with increasing OTA concentration (Figure 1), which is consistent with previous studies demonstrating that OTA can induce apoptosis of intestinal cells [20,44,45]. Apoptosis, which is a form of programmed cell death, may disrupt the integrity of the intestinal barrier [46,47]. OTA induces apoptosis by modulating BcL-2 family members [30,48], and activation of MEK/ERK1-2 signaling has been shown to be crucial for OTA-induced apoptosis in HK-2 cells [49]. Although OTA-induced apoptosis has been studied previously, there has been a tendency to focus mainly on a particular pathway [20,49,50]. In this study, we chose instead to search for related pathways from a holistic perspective because transcriptome data can quickly and economically supply accurate genome information, particularly in identifying the effects of biological pathways and processes [51,52]. Elena et al. (2016) used transcriptomic analysis to reveal the different toxicity mechanisms of OTA and citrinin [53]. Hence, we used a transcriptomic study to provide a genome-wide biological view of changes that occurred after treatment of differentiated Caco-2 cells with different concentrations of OTA.

For the GO annotation, the main GO terms of the DEGs were cell processes and metabolic processes. Another microarray study demonstrated that OTA decreased mRNA levels of genes involved in liver cell metabolism [54]. KEGG enrichment analysis showed that there are key pathways, such as the cell cycle, the MAPK signaling pathway, the TNF signaling pathway, and the p53 signaling pathway, that are highly associated with cell apoptosis (Figure 4). These results indicate that OTA-induced cell apoptosis probably involves changes to the enriched pathways, such as the cell cycle, and other important signaling pathways. From the results obtained in this study, we speculate that cell apoptosis happens in an orderly manner and can be regulated by many mechanisms. Liang et al. [55] pointed out that OTA suppresses the cell cycle, particularly DNA replication, leading to cell cycle arrest and apoptosis. OTA has been shown to regulate apoptosis, the cell cycle, and cell fate following DNA damage by stimulating MAPK signaling pathways, such as p38 MAPK, ERK1/2, and JNK [56,57,58]. Apoptosis of liver and/or kidney cells has also been shown to be influenced by activation of certain signal transduction pathways, such as MAPK, ERK, p38, and JNK [56,59,60]. Another study obtained data that were of great importance in characterizing the modulatory effect of the TNF signaling pathway on apoptotic signaling [61]. Activation of the p53 pathway has also been shown to play a key role in OTA-induced apoptosis of human and monkey kidney cells [33]. These results are in agreement with reported research demonstrating that OTA triggers a p53-dependent apoptotic pathway in human hepatoma cells [62]. Using knockout mice, Hibi et al. showed that the p53 pathway plays a key role in OTA-induced genotoxicity [63]. Because previous research has highlighted the role of p53 signaling in OTA-induced apoptosis, in this study, we concentrated on the p53 pathway and analyzed its specific regulatory mechanism.

The p53 signaling pathway, which is amplified by different concentrations of OTA, plays a key role in regulating DNA repair mechanisms, oxidative stress, cell cycle arrest, and cell apoptosis [64,65]. In the renal outer medulla, OTA causes genotoxicity by deregulating molecular functions such as DNA double-strand break repair, cell cycle arrest in response to DNA damage, and p53-associated factors [66]. Alterations in oxidative stress and calcium homeostasis have also been associated with OTA-induced toxicity [67]. The proto-oncogene, MDM2, a nuclear protein that is the main cellular antagonist of p53, can regulate the potentially lethal activities of p53 by tightly combining with a p53 tumor suppressor protein and negatively regulating its stability and transcriptional activity [68]. Furthermore, MDM2 can not only inhibit the transactivation of p53 activity, but also stabilize p53 and reduce p53-mediated apoptosis [69]. The p53-MDM2 feedback loop may be important in regulating cell apoptosis. Previous studies have demonstrated that apoptotic pathways are regulated by proteins such as the tumor suppressor p53 and inhibitor of apoptosis (IAP) proteins, which are highly regulated by MDM2 through an autoregulatory feedback loop [70,71]. OTA may induce apoptosis in differentiated Caco-2 cells through the feedback loop interaction of MDM2 and the p53 signaling pathway. A study by Gu et al. [72] showed that gambogic acid induced apoptosis of wild-type p53-expressing cancer cells through downregulation of MDM2. Another study found that violacein decreased the expression of MDM2 and caused apoptosis of human breast cancer cells by activating PARP, CDKN1A, TNF-α, and p53 cleavage [73]. Our transcriptome analysis showed that OTA induced apoptosis in differentiated Caco-2 cells by activating key downstream genes in the p53 signaling pathway. In these key pathways, some DEGs, such as *CASP3*, *AKT1*, *MAPK1*, *HRAS*, *PIK3CA*, and *MDM2*, were highly involved (Figure 5), which appeared in the group 4 µg/mL specially instead of common DEGs (Figure 3). Several genes (*CASP3*, *CDC25B*, *CXCL*, and *EGR1*) were randomly selected from these key DEGs and common DEGs (*FRYL*, *TFEC*, *SHPK*, and *H2BFS*) were used for gene verification by qRT-PCR analysis. These key genes, and other DEGs, showed similar results in the RNA-seq analysis (Figure 6), and further study is needed to elucidate the relationships between the genes and regulation of cell apoptosis. In contrast to our own study, Kuroda et al. found that expression of the *CDK1* gene was increased by treatment with OTA [74]. We believe that the difference between the studies is because OTA increases DNA damage in *gpt* delta rats and CDK1 could accelerate the resection of DNA broken ends during the homologous recombination process [74,75,76]. Through analysis of the enrichment pathway, we found that p53 is a target signaling pathway and speculate that OTA probably induces cell apoptosis by downregulating MDM2 expression. Western blotting confirmed that OTA-induced cell apoptosis in differentiated Caco-2 cells is linked to perturbation of MDM2 (Figure 7).

In addition to the downregulation of MDM2, there are several important genes in the p53 signaling pathway that are upregulated in the apoptosis of differentiated Caco-2 cells. These are Noxa (phorbol-12-myristate-13-acetate-induced Protein1, PMAIP1), tumor protein p53 regulated apoptosis-inducing protein 1 (P53AIP1), and caspase 3 (cysteinyl aspartate proteases, CASP3). P53 transactivates noxa, bax, puma, and other apoptotic response genes, and their products trigger mitochondrial apoptosis pathways [50]. Noxa is a member of the pro-apoptotic B-cell lymphoma 2 (BCL-2) family and can not only promote activation of caspases and apoptosis, but also promote changes in the mitochondrial membrane and outflow of apoptogenic proteins from the mitochondria [77,78]. Functional studies have shown that P53AIP is a pro-apoptosis molecule, and it is believed to play a vital role in mediating p53-dependent apoptosis. P53AIP1 can lead to CASP3 activation and downstream activities by decreasing mitochondrial membrane potential and inducing release of cytochrome c. CASP3 plays a vital role in many events involved in cell apoptosis, including activation of the caspase cascade and the execution of apoptosis [79,80,81]. OTA-induced downregulation of MDM2 could regulate the downstream upregulation of Noxa and P53AIP1, and eventually activate CASP3 in the apoptosis of Caco-2 cells.

To conclude, we have shown that OTA-induced apoptosis in differentiated Caco-2 cells may involve downregulation of MDM2 and upregulation of CASP3, through activation of the p53-mediated cell apoptosis signaling pathway. A more exhaustive study is, however, needed to fully elucidate all of the underlying mechanisms. We have also demonstrated the intestinal toxicity of OTA and provided a genome-wide view of biological responses, which provides a theoretical basis for enterotoxicity and should be useful in establishing an MRL for OTA.

## 4. Materials and Methods

### 4.1. Chemicals and Reagents

OTA powder (C_20_H_18_ClNO_6_; molecular weight, 403) was purchased from Pribolab (Qingdao, China). Human colon adenocarcinoma Caco-2 cells (passage number 18) were acquired from the American Type Culture Collection (Manassas, VA, USA). Fetal bovine serum (FBS) and Dulbecco’s modified Eagle’s medium (DMEM) were obtained from Gibco (Carlsbad, CA, USA). Nonessential amino acids (NEAA), trypsin (2.5%), antibiotics (100 units/mL penicillin, 100 µg/mL streptomycin), phosphate-buffered saline (PBS), Western and IP Cell Lysis Buffer, and an Annexin V-FITC apoptosis detection kit were supplied by Beyotime Biotechnology (Shanghai, China). A stock solution of OTA (1000 μg/mL) was obtained by dissolving OTA in methanol, and stored at −20 °C for later use. Rabbit anti-β-actin (58169S) and rabbit anti-MDM2 antibodies (86934S) were obtained from Cell Signaling Technology (Boston, MA, USA), and goat anti-rabbit IgG conjugated to horseradish peroxidase was obtained from Bioss (Beijing, China).

### 4.2. Cell Culture and Treatments

Differentiated Caco-2 cells were seeded into six-well Transwell chambers (Corning, NY, USA) at a density of 1 × 10^5^ cells per well in DMEM containing 4.5 g/L glucose, 1% NEAA, 10% FBS, and antibiotics. The cells were incubated at 37 °C in a humidified atmosphere containing 5% CO_2_. A polarized epithelial monolayer of differentiated Caco-2 cells was formed by replacing the medium every other day for 21 days. Because an MRL for OTA in milk has not yet been established and previous studies have indicated that OTA and aflatoxin M1 (AFM1) have the same cytotoxicity in human intestinal Caco-2 cells [82], we chose the concentration of 0.0005 μg/mL for OTA, based on the Chinese (0.5 μg/kg) limits for AFM1 in milk. We established a concentration gradient when measuring cell viability and transepithelial electrical resistance [83] and chose the 4 μg/mL concentration to investigate the effect of an upper concentration limit on intestinal injury, without affecting cell survival. After 48 h of treatment with OTA (0.0005, 0.005, and 4 μg/mL), serum-free medium containing the same concentration of methanol was added to the control group. Differentiated Caco-2 cells were collected after 48 h for subsequent cell apoptosis studies, transcriptomics analysis, and qRT-PCR.

### 4.3. Cell Apoptosis Assay by Annexin V-FITC/PI FACS

Following the instructions of the Annexin V-FITC apoptosis detection kit, the differentiated Caco-2 cells were washed twice with cold PBS after treatment with different concentrations of OTA for 48 h. Cell samples were trypsinized, transferred to centrifuge tubes, and centrifuged at 1000× *g* for 5 min. The supernatant was discarded and the cells were resuspended in PBS (1 mL). The cell suspension was transferred to a 1.5 mL centrifuge tube and centrifuged again at 1000× *g* for 5 min. The supernatant was discarded and the cells were gently resuspended by adding Annexin V-FITC (5 μL), propidium iodide (10 μL), and Annexin V-FITC binding solution (195 μL). The cells were gently vortexed and incubated for 10–20 min at room temperature. An FC 500 MCL flow cytometer (Becton Dickinson, Mountain View, CA, USA) was used to analyze the cell samples within 1 h.

### 4.4. RNA Extraction, Library Construction, and Transcriptome Sequencing

#### 4.4.1. RNA Extraction

After treatment with OTA (0.0005, 0.005, and 4 μg/mL) for 48 h, Trizol (Invitrogen, Camarillo, CA, USA) was used to extract total RNA from the differentiated Caco-2 cells in accordance with the manufacturer’s instructions. Remaining DNA was then removed by treatment with RNase-free DNase I (Takara Bio, Kusatsu, Shiga, Japan) for 30 min at 37 °C. Triple replicates of each treatment of differentiated Caco-2 cells were combined into a sample to reduce sample variability. After purification using an RNeasy Mini Kit (Qiagen, Dusseldorf, Germany), a NanoDrop 2000 spectrophotometer (Thermo Scientific, Wilmington, DE, USA) was used to assess RNA quality and RNase-free agarose gel electrophoresis was used to assess quantity.

#### 4.4.2. Construction of cDNA Library and Illumina Sequencing

RNA-seq was performed on RNA samples from differentiated Caco-2 cells treated with each of the three concentrations of OTA described above. High-quality RNA from each sample was enriched using magnetic Oligo (dT) beads and used to construct and sequence a cDNA library. Suitable cDNA fragments were selected as templates for PCR amplification using index primers and NEB universal PCR primers. When constructed, the cDNA library was sequenced using a HiSeq^TM^ 2500 RNA sequencing system (Illumina, San Diego, CA, USA).

#### 4.4.3. De Novo Assembly and Quantification of Gene Abundance

Before data analysis, the quality of the original data was controlled and noise was reduced by data filtering. In order to obtain high-quality clean reads for subsequent information analysis, the clean reads were filtered more rigorously to remove reads containing adapters or more than 10% of unknown nucleotides (N), as well as all reads with more than 50% low-quality sequence (*q*-value ≤ 20). The transcriptome assembly program Trinity was used for de novo transcriptome reconstruction. The sequencing error rate was used to evaluate the quality of sequenced reads, the saturation of the library, and the randomness of sequencing. Cuffmerge was used to combine the transcripts from different replicates of one group into a comprehensive set of transcripts for subsequent differential analysis and to filter the unique annotation files for manually introduced assembly errors. Annotation of unigenes was obtained using Blast and compared with data obtained using GO, KEGG, KOG, NR, COG, and Swissprot databases. Gene expression levels were then calculated using both the “raw counts” mode and “FPKM” [84].

#### 4.4.4. Differentially Expressed Genes and their Dynamic Expression Profile

The edge R package (http://www.rproject.org/) was used to determine the DEGs between different treatment groups [62]. Firstly, we used the general filtering standard (FDR (False Discovery Rate) < 0.05 and |log2FC|>1) to identify significant DEGs. All DEGs were analyzed using the short time-series expression miner [85]. GO annotation was analyzed using Blast2GO software [86]. Blast software was used against the KEGG database to determine the KEGG pathway annotation [75]. GO/KEGG functional enrichment analysis was carried out for the genes in each trend, and the *p*-value was calculated by the hypothesis test. After the *p*-value was corrected by FDR, the GO term and path satisfying this condition were assigned a *q*-value ≤ 0.05 threshold.

### 4.5. Validation of RNA-Seq Results Using qRT-PCR

A Prime Script™ II 1st Strand cDNA Synthesis Kit and TB Green™ Premix Ex Taq™ II (Takara, Kusatsu, Shiga, Japan) were used to reverse transcribe cDNAs. The qRT-PCR conditions were as follows: 95 °C for 30 s; 40 cycles at 95 °C for 5 s; 60 °C for 30 s; and, finally, 72 °C for 20 s. The sequences of the primers, which were synthesized by a commercial company (Sangon Biotech Co., Ltd., Shanghai, China), are shown in Table 2. The data were analyzed and 7500 Software v.2.0.1 (Applied Biosystems, Foster City, CA, USA) was then used to determine the cycle threshold (Ct). The GAPDH gene (house-keeping gene) was selected as an internal control to normalize the expression data. The 2^−ΔΔCt^ method was used to calculate the relative expression of genes, and the mean and standard deviation of three biologic replicates are shown as the results [87].

### 4.6. Western Blotting Assays

Western blotting was used to verify differentially expressed proteins. Caco-2 cells were cultured in transwell chambers, with or without OTA (0.0005, 0.005, and 4 μg/mL), for 48 h. First, the cell samples were lysed with Western and IP Cell Lysis Buffer, and then equal amounts of protein were subjected to sodium dodecyl sulfate polyacrylamide gel electrophoresis (SDS-PAGE). The samples were next transferred to polyvinylidene fluoride membranes by dry rotation and the membranes were blocked with 5% skim milk powder dissolved in Tris-buffered saline for 2 h at room temperature. The samples were then incubated for 2 h with specific primary antibodies (1:1000 in TBS), washed three times with TBST (TBS containing 0.1% Tween 20), and incubated for 2 h with secondary antibodies. The bands were imaged using a Tanon-5200 Chemiluminescent Imaging System (Tanon Science & Technology Co., Ltd.) and band densities were analyzed using Image J 2 × software (Version 2.1.0, National Institutes of Health, Bethesda, MD, USA, 2006). Intensity values were normalized to human β-actin.

### 4.7. Statistical Analysis

Analysis of data was performed using SPSS^®^ Statistics version 19. Analysis of variance (ANOVA) followed by Tukey’s multiple comparison was used to test statistical differences between the OTA treatment groups and the control group. An asterisk (*) indicates a *p*-value < 0.05, which was regarded as statistically significant.

## Figures and Tables

**Figure 1 toxins-12-00023-f001:**
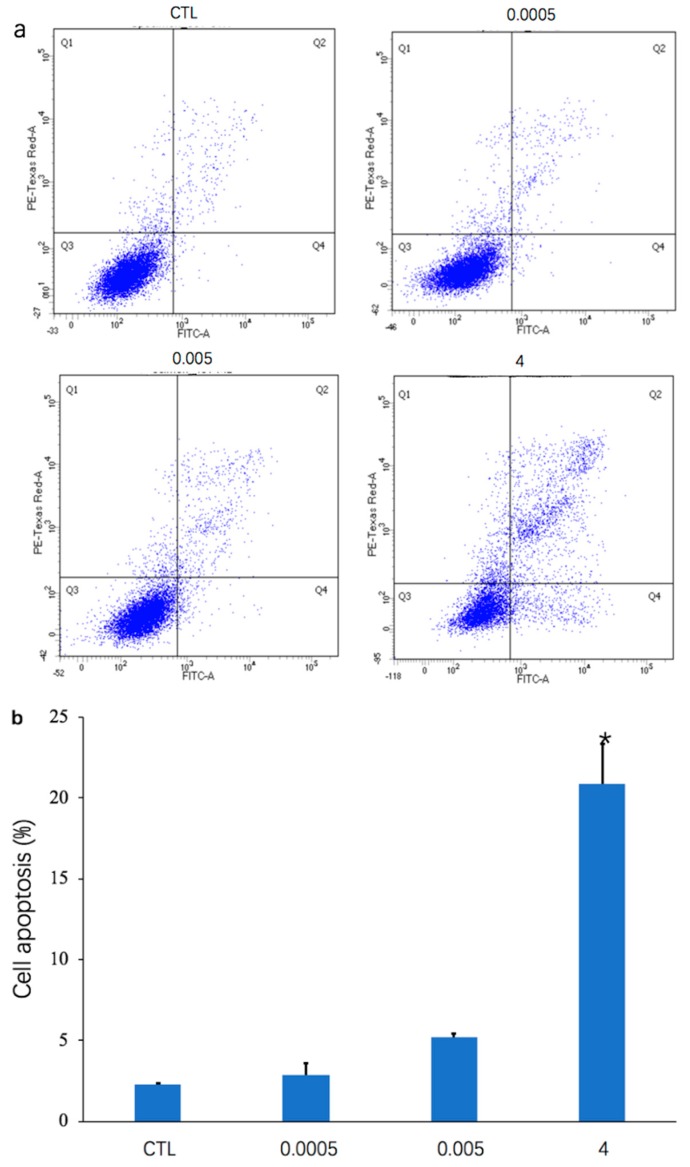
Effects of ochratoxin A (OTA) on apoptosis in differentiated Caco-2 cells. Incubation of cells for 48 h with OTA (0, 0.0005, 0.005, and 4 μg/mL) was followed by analysis using flow cytometry. (**a**) Representative flow cytometry plots are presented for the control group (CTL) and OTA groups (0.0005 μg/mL, 0.005 μg/mL, and 4 μg/mL). (**b**) Histogram showing number of differentially expressed genes after treatment with three concentrations of OTA compared with the control group. * represents a significant difference (*p* < 0.05).

**Figure 2 toxins-12-00023-f002:**
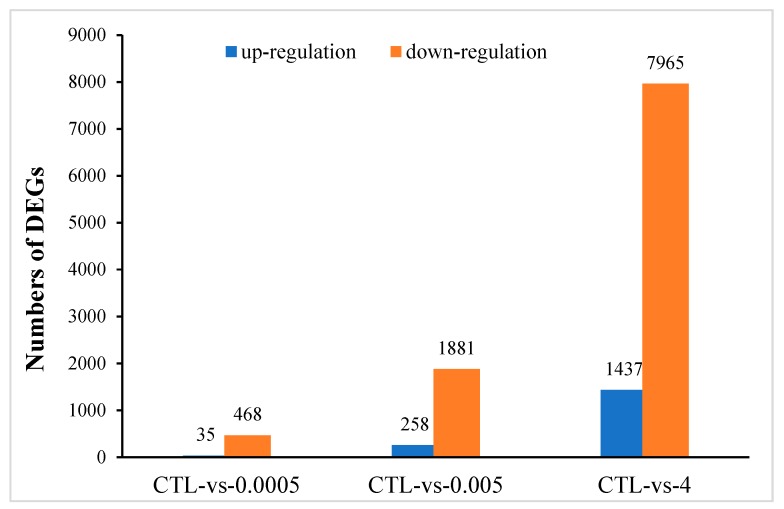
Differentially expressed genes (DEGs) identified in differentiated Caco-2 cells after exposure to 0.0005, 0.005, and 4 μg/mL OTA for 48 h. Histogram shows the number of DEGs compared with the control (CTL).

**Figure 3 toxins-12-00023-f003:**
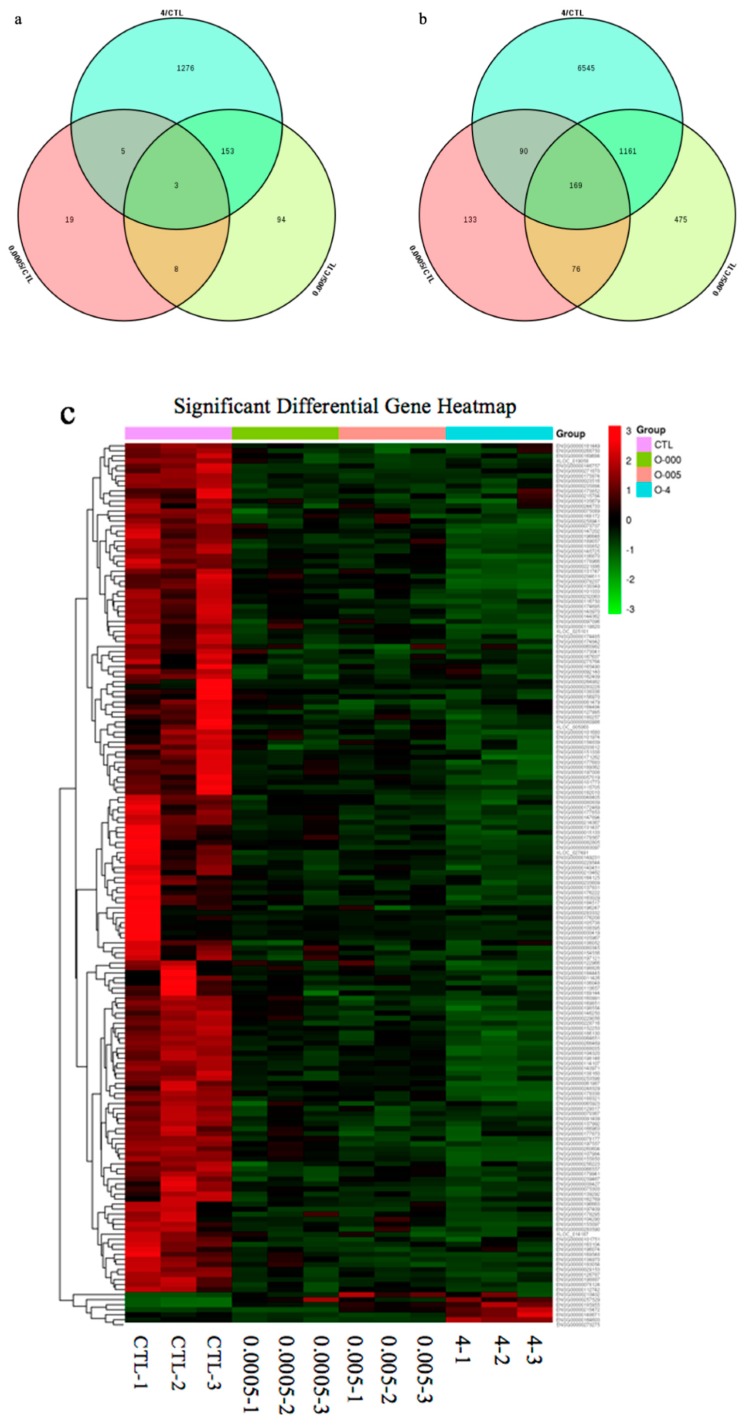
Gene expression profiles of differentiated Caco-2 cells treated with ochratoxin A (OTA). (**a**) Venn diagram depicting upregulated DEGs common to different doses of OTA. (**b**) Venn diagram depicting downregulated DEGs common to different doses of OTA. (**c**) Hierarchical clustering of common DEGs in differentiated Caco-2 cells based on log10-transformed expression values (fragments per kilobase of transcript per million fragments mapped, FPKM).

**Figure 4 toxins-12-00023-f004:**
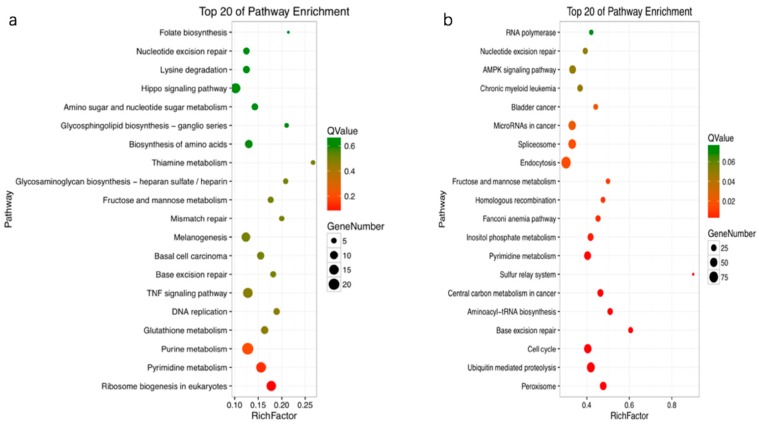
Scatter plots of the top 20 enriched Kyoto Encyclopedia of Genes and Genomes (KEGG) pathway terms (**a**), profile 0; (**b**), profile 3. Enriched items were measured by the rich factor, q value (*q* < 0.05), and number of genes. TNF, tumor necrosis factor.

**Figure 5 toxins-12-00023-f005:**
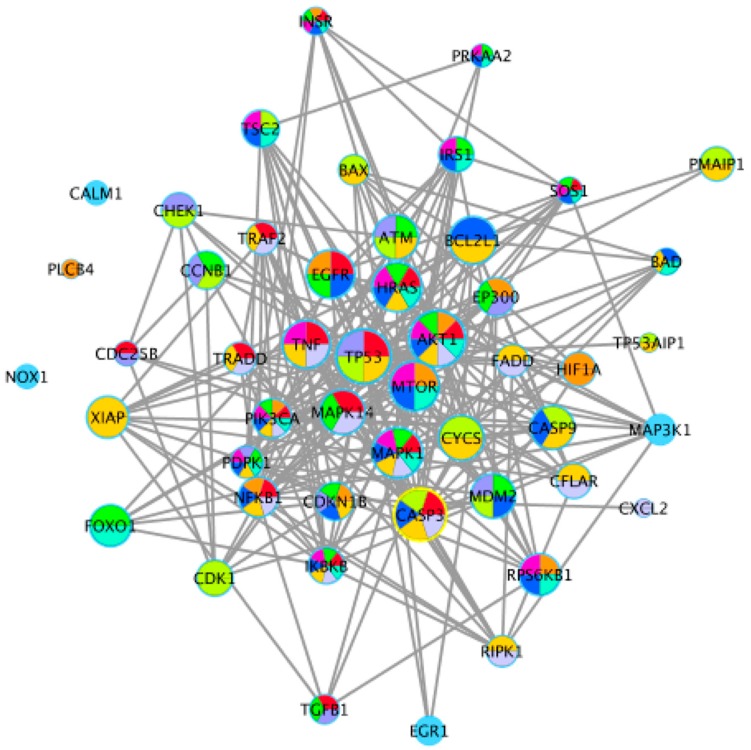
Protein–protein interaction (PPI) network for 50 important enzymes encoded by key differentially expressed genes (DEGs) of key pathways. Different nodes represent different enzymes. Interactions between these enzymes are represented by different size nodes; the larger the node, the stronger the connectivity.

**Figure 6 toxins-12-00023-f006:**
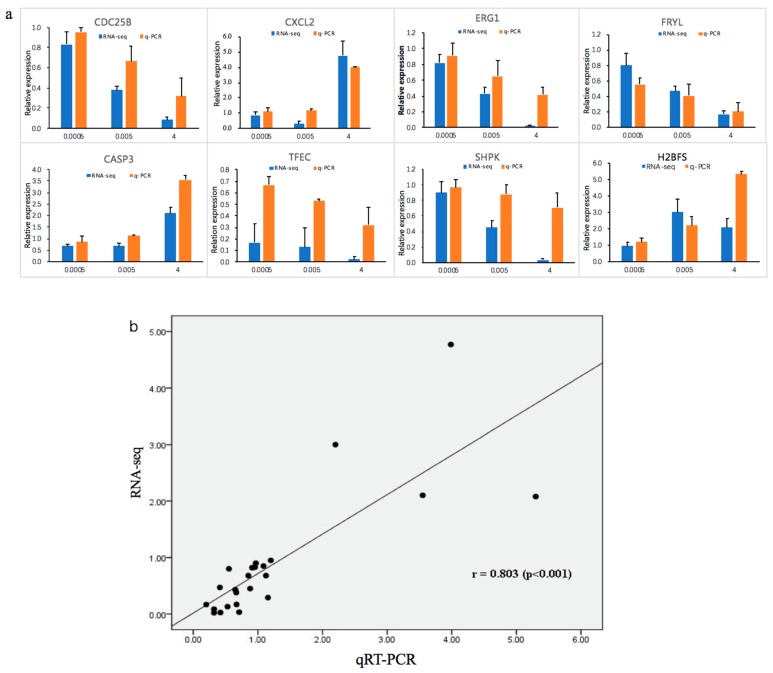
(**a**) Quantitative reverse transcription-polymerase chain reaction (qRT-PCR) results for eight DEGs compared with RNA-seq results. The blue bars represent RNA-seq data and the orange bars represent q-PCR data. (**b**) Correlation analysis between RNA-seq and qPCR.

**Figure 7 toxins-12-00023-f007:**
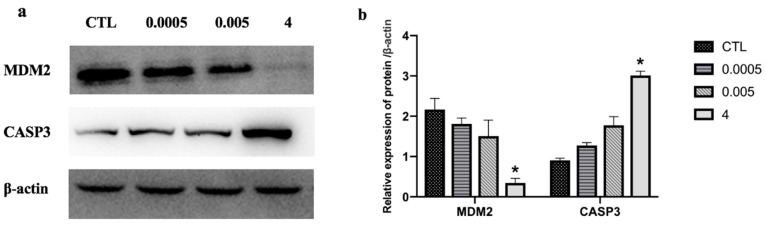
Effect of OTA (0.0005, 0.005, and 4 μg/mL) on cell apoptosis-related proteins in differentiated Caco-2 cells. Protein extracts were immunoblotted for candidate proteins (**a**) and band densities were quantified (**b**). Results are the mean of three separate experiments performed in triplicate ± SD. * *p* < 0.05, significantly different from control group.

**Table 1 toxins-12-00023-t001:** Key pathways and important related differentially expressed genes (DEGs) that are associated with the toxic effect of ochratoxin A (OTA) in differentiated Caco-2 cells. mTOR, mechanistic target of rapamycin kinase; MAPK, mitogen-activated protein kinase; TNF, tumor necrosis factor.

Pathway	Pathway ID	Key DEGs from the Pathway
Cell cycle	ko04110	MDM2, CDK1, TP53, EP300, ATM, CDKN1B, TGFB1, CHEK1, CDC25B, CCNB1, PDPK1
P53 signaling pathway	ko04115	MDM2, PMAIP1, CASP3, TP53AIP, CDK1, TP53, ATM, CYCS, TSC2, CHEK1, CASP9, CCNB1, BAX
HIF-1 signaling pathway	ko04066	HIF1A, NFKB1, CDKN1B, MTOR, EP300, AKT1, PIK3R, EGFR, NOX1, INSR, PIK3CA
PI3k-Akt signaling pathway	ko04151	CDKN1B, NFKB1, CCND1, BCL2L1, MYC, PIK3CA, EGFR, MDM2, TP53, IKBKB, AKT1, HRAS, TSC2, IL3RA, MAPK1, SOS1, CASP3, MTOR, INSR, BAD, CASP9, PDPK1
mTOR signaling pathway	ko04150	MTOR, TNF, IKBKB, PRKAA, PIK3CA, HRAS, TSC2, MAPK1, AKT1, SOS1, INSR, PDPK1
Apoptosis	ko04210	XIAP, BCL2L1, NFKB1, PIK3CA, CASP3, CYC, BAX, BAD, IL3RA, CASP9, CFLAR, TP53, ATM, HRAS, TRADD, MAPK1, RIPK1, CDKN1B, PMAIP1, TP53AIP, AKT1, TNF, IKBKB
Foxo signaling pathway	ko04068	CDKN1B, PIK3CA, MDM2, PDPK1, EGFR1, FOXO1, AKT1, IRS1, INSR, MDM2, TGFB1, IKBKB, ATM, HRAS, MAPK1, SOS1, EGFR, EP300, CCNB1, TGFB1
Insulin signaling pathway	ko04910	MAPK1, SOS1, PIK3CA, AKT1, MTOR, IRS1, INSR, HRAS, TSC2, TRADD, MAPK1, CALM1, IKBKB, FOXO1
MAPK signaling pathway	ko04010	TNF, CASP3, NFKB1, TP53, MAPK14, TGFB1, PIK3CA, HRAS, SOS1, EGFR, MAPK1, TRAF2, CDC25B, AKT1, TNF, IKBKB
TNF signaling pathway	ko04668	XIAP, TNF, NFKB1, CASP3, IKBKB, MAPK1, MAPK14, CASP8, TRADD, FADD, MAPK1, RIPK1, TRAF2, AKT1, PIK3CA

**Table 2 toxins-12-00023-t002:** Primer sequences for the quantification of genes by quantitative reverse transcription-polymerase chain reaction (qRT-PCR). CASP 3, caspase 3; CXCL2, C–X–C motif chemokine ligand 2; CDC25B, cell division cycle 25B; EGR1, early growth response 1 (EGR1), H2BFS, H2B histone family members; SHPK, sedoheptulokinase; TFEC, transcription factor EC.

Genes	Product Length (bp)	Forward Primer Sequence (5′–3′)	Reverse Primer Sequence (5′–3′)
GAPDH	235	GGAGTCCACTGGCGTCTT	GAGTCCTTCCACGATACCAAA
CASP3	109	TCCTGAGATGGGTTTATGT	TGTTTCCCTGAGGTTTGC
CXCL2	150	CCAAACCGAAGTCATAGC	GAACAGCCACCAATAAGC
CDC25B	296	GTAGACGGAAAGCACCAAGA	TCCCTGATGAAACGGCAC
EGR1	229	CACGAACGCCCTTACGCT	CATCGCTCCTGGCAAACT
H2BFS	119	TGCTCGTCTCAGGCTCGTAG	CTTCCTGCCGTCCTTCTTCT
SHPK	58	AGTAGATGCGGCAATGGT	TTGGTAGGGATGGCTGTG
TEFC	94	GCACTGGAGGGATAAATG	TAAAGACACCCGAAGGAT

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
