# Peer review of "Transcriptome Analysis of Ochratoxin A-Induced Apoptosis in Differentiated Caco-2 Cells"

_toxins, 2019, doi:10.3390/toxins12010023_

Round 1
Reviewer 1 Report
In this study, the authors assessed OTA effects on differentiated intestinal Caco-2 cells through transcriptomic analysis. 10 key pathways and 24 key DEGs, were identified, as a potential target for the selection of candidate genes for cell apoptosis. These results were confirmed by qRT-PCR analysis revealing the involvement of CASP3 in the regulation of the apoptosis pathway. This study provides a genome-wide biological response view of the OTA intestinal toxicity.
Although high impact techniques were used in this research study, in many parts I could not understand what the authors meant. Extensive English editing is required throughout the whole manuscript. In addition, I have the following comments:
How did the authors select OTA concentrations? lines 64-68 check for plagiarism.Author Response
Please see the attachment

Reviewer 2 Report
The paper describes the impact of different OTA concentrations mostly on the transcriptome of human epithelial colorectal carcinoma Caco-2 cells. The authors apply 3 different OTA concentrations and determine the transcriptome after 48 hours of exposure. Mostly down-regulated genes are found by RNA seq, which are then clustered into functional GO groups and investigated for protein-protein networks and common signal transduction pathways affected by the mycotoxin. The authors finally corroborate the RNA seq data at some up- or down-regulated loci by standard RT-PCR and for two divergently regulated proteins by western blot. The work is entirely descriptive and can serve as a valuable source of genomic data for OTA affected gene functions in order to clarify the pathways involved in OTA toxicity in the future.
Before publications, the authors should consider the following specific suggestions:
1) Fig. 3A represents both up- and down-regulated gene functions after OTA treatment. However, it would be more informative to analyze the differentially affected gene functions separately to eventually point out up-regulated defense/cell death induction functions separately from down-regulated potential OTA targets.
2) Surprisingly, the authors find relatively little overlap, which means commonly affected genes, by the three different OTA doses. I would have expected largely overlapping gene clusters, which might increase with dose. The authors should comment on these results.
3) To this reviewer, Figure 4 is not very meaningful and might actually be removed. It contains numerous GO groups which are not contributing to shed light on OTA function, such as "cellular process", "metabolic process", "cell part" etc
4) The authors must comment on previously published transcriptomic studies performed with OTA in other cell systems:
Hibi D, Kijima A, Kuroda K, Suzuki Y, Ishii Y, Jin M, Nakajima M, Sugita-Konishi Y, Yanai T, Nohmi T, Nishikawa A, Umemura T. Molecular mechanisms underlying ochratoxin A-induced genotoxicity: global gene expression analysis suggests induction of DNA double-strand breaks and cell cycle progression. J Toxicol Sci. 2013 Feb;38(1):57-69. Hundhausen C, Boesch-Saadatmandi C, Matzner N, Lang F, Blank R, Wolffram S, Blaschek W, Rimbach G. Ochratoxin a lowers mRNA levels of genes encoding for key proteins of liver cell metabolism. Cancer Genomics Proteomics. 2008 Nov-Dec;5(6):319-32. Arbillaga L, Vettorazzi A, Gil AG, van Delft JH, García-Jalón JA, López de Cerain A. Gene expression changes induced by ochratoxin A in renal and hepatic tissues of male F344 rat after oral repeated administration. Toxicol Appl Pharmacol. 2008 Jul 15;230(2):197-207. Vanacloig-Pedros E, Proft M, Pascual-Ahuir A. Different Toxicity Mechanisms for Citrinin and Ochratoxin A Revealed by Transcriptomic Analysis in Yeast. Toxins (Basel). 2016 Sep 22;8(10).Author Response
Please see the attachment

Reviewer 3 Report
line 17 – I would suggest: "OTA may impair intestinal function" – in accordance with the holistic approach;
lines 19-21 – This is a better summary and conclusion than that at the end of the Discussion section;
lines 78-80; 124-127; 137-142; 171-172; 188-190 – In my opinion, the values should not be analyzed in the Results section. Interpretations and analyses should be included in the Discussion section;
lines 277-284 – This is not a summary/conclusion. The Authors have repeated the results in a more condensed manner, but conclusions or suggestions have not been formulated;
lines 284-286 – What are the implications?
